# Simulation and Experimental Validation of a Pressurized Filling Method for Neutron Absorption Grating

**DOI:** 10.3390/mi14051016

**Published:** 2023-05-09

**Authors:** Eryong Han, Kuanqiang Zhang, Lijuan Chen, Chenfei Guo, Ying Xiong, Yong Guan, Yangchao Tian, Gang Liu

**Affiliations:** National Synchrotron Radiation Laboratory, University of Science and Technology of China, Hefei 230029, China; haneryong@mail.ustc.edu.cn (E.H.);

**Keywords:** neutron absorption grating, particle filling method, pressurized filling, filling rate

## Abstract

The absorption grating is a critical component of neutron phase contrast imaging technology, and its quality directly influences the sensitivity of the imaging system. Gadolinium (Gd) is a preferred neutron absorption material due to its high absorption coefficient, but its use in micro-nanofabrication poses significant challenges. In this study, we employed the particle filling method to fabricate neutron absorption gratings, and a pressurized filling method was introduced to enhance the filling rate. The filling rate was determined by the pressure on the surface of the particles, and the results demonstrate that the pressurized filling method can significantly increase the filling rate. Meanwhile, we investigated the effects of different pressures, groove widths, and Young’s modulus of the material on the particle filling rate through simulations. The results indicate that higher pressure and wider grating grooves lead to a significant increase in particle filling rate, and the pressurized filling method can be utilized to fabricate large-size grating and produce uniformly filled absorption gratings. To further improve the efficiency of the pressurized filling method, we proposed a process optimization approach, resulting in a significant improvement in the fabrication efficiency.

## 1. Introduction

Neutron imaging has emerged as a powerful nondestructive inspection technique for detecting the interior of metallic materials owing to the high penetration ability of neutrons [1,2,3]. Neutron phase contrast imaging has been developed for materials that exhibit low attenuation to neutron beams, thereby allowing for higher resolution, higher image contrast, and higher measurement sensitivity [4,5]. The Talbot Lau interferometer, which employs three gratings, is the most common approach used for neutron phase contrast imaging. The absorption grating (G_0_) generates spatially coherent neutron beamlets, which are phase shifted by the second grating (G_1_) to produce transverse intensity modulation. However, this spatial modulation is too small to be resolved by the current neutron detector. Therefore, a third absorption grating (G_2_) should be used to analyze the signal [6,7,8].

Despite its potential, the fabrication of neutron absorption gratings remains a key challenge in this technology. Gd is the preferred material for the absorber of neutron absorption gratings due to its high neutron absorption coefficient [9,10,11]. For strong absorption (97%), approximately 11 µm of Gd is required for a neutron wavelength of 4.1 Å or higher [12,13]. However, Gd is a very rare material in micro-nanofabrication and there is no universal deposition process [4]. Consequently, various fabrication processes have been developed to achieve this goal for different requirements, such as particle filling methods [9,13], the evaporation of pure gadolinium metal [10,11,12,14,15], and the embossing of gadolinium-alloyed metallic glasses [16,17,18], each of which has its own characteristics and applicability.

The particle filling method is advantageous for making large-size gratings, which can provide a large imaging field of view [4,19]. However, during particle deposition, the pores between the particles are large, resulting in a low particle packing density. This leads to a low particle filling rate and the required absorber height being greater than the design height [9,13]. Accordingly, an increase in the aspect ratio of the grating will increase the difficulty of fabrication. In the fabrication of the silicon grating structure, although current processes can achieve high aspect ratio structures (>50:1) [20], there are nonuniformity issues when fabricating large-size gratings. Moreover, when filling particles into the grating structure to fabricate an absorption grating, particle filling becomes difficult as the aspect ratio of the grating increases, which also poses the issue of nonuniformity. Therefore, it is necessary to increase the filling rate of the particles to reduce the aspect ratio of the grating. Ultrasound and high-speed centrifugation are commonly employed to address the low particle filling rate [21,22,23]. However, the grating structure may be damaged during ultrasonication, and the high-speed centrifuge is only suitable for making small-size gratings, which will lead to uneven particle filling for making large-size absorption gratings. In this paper, we utilize the pressurized filling method to enhance the particle filling rate and explore the factors influencing the rate during pressurized filling through simulation. Additionally, we propose a method for optimizing the particle filling process to improve the fabrication efficiency of absorption gratings.

## 2. Experimental Methods and Simulation

### 2.1. Methods

The filling rate is related to the particle size and particle arrangement. When the particle size is fixed, the particle arrangement can be changed to improve the particle filling rate. Particle rearrangement occurs when the particles are subjected to external forces during the stacking process. Ultrasound is used to uniformly distribute particles using vibration, while high-speed centrifugation is used to redistribute particles using centrifugal force. In this paper, external pressure was used to rearrange the particles as shown in Figure 1. The particles were deposited into the grating grooves, and then a soft material was placed on the surface of the grating structure, followed by uniform pressure to deform the material downward into the grating grooves. The particles in the grooves were compressed by the material and moved downward, causing their rearrangement and an increase in the particle stacking density. Then, we continued to deposit the particles into the grooves of the grating, applying pressure to the surface of the material so that the accumulation density of the particles in the grooves continued to increase. We repeated the pressure filling and gradually increased the filling rate.

### 2.2. Experiment

A neutron absorption grating was fabricated using the pressurized filling method, as depicted in Figure 1. To create a neutron absorption grating, a silicon grating structure was initially fabricated using a wet etching method with a KOH solution. To prepare a particle suspension, gadolinium oxide (Gd_2_O_3_, Zhongnuo Advanced Materials (Beijing, China) Technology Co., Ltd.) particles with a size range of 1–3 μm were sonicated with ethanol at a mass ratio of 1:100, using an ultrasonic machine (XZ-3DTD, Ningbo Advanced Leaf Biological Technology Co., Ltd., Ningbo, China) at a power of 750 W for 20 min. This suspension was poured onto the surface of the silicon wafer and allowed to settle for 3 h. Once the ethanol had completely evaporated, the Gd_2_O_3_ particles were naturally deposited and filled the grooves of the grating. The remaining Gd_2_O_3_ particles on the surface of the silicon structure were then removed. Subsequently, a soft material such as PDMS (polydimethylsiloxane, Dow Corning, Midland, TX, USA) with Young’s modulus of 7.5 × 10^5^ Pa was placed on the surface of the grating, using an embossing device (ZKRYM-1, Institute of Optics and Electronics, CAS, Chengdu, China) with a pressure head size of 40 cm × 40 cm to ensure that the pressure head was much larger than the pressed material so that the pressure was applied uniformly onto the PDMS surface. After releasing the pressure, the suspension was used to continue depositing particles into the grating grooves. Pressure was reapplied repeatedly to gradually increase the filling rate. The grating structures and experimental parameters are shown in Table 1.

### 2.3. Simulation

To investigate the pressurized filling method, COMSOL Multiphysics was used to simulate the material deformation during the particle filling process. As shown in Figure 2a, the model consisted of silicon, PDMS, and porous particle models. The deformation of the material in the grating groove was simulated when pressure was applied to the material on the grating surface. The mass of the particles in the grating groove remained unchanged after pressurization, and the volume of the particle model decreased. Using the particle mass divided by the total mass of fully filled gadolinium oxide material at the current volume, we obtained the particle packing density after each pressurization, which resulted in the corresponding particle filling rate. The filling rate of the particles in the grating groove in the simulation and experiment is calculated as
(1)$$D=\frac{M}{{(V}_{1}-{\;V}_{2})\rho }$$
where *D* is the filling rate, *M* is the mass of particles filled into the grating tank before pressurization, *ρ* is the density of Gd_2_O_3_, *V*_1_ is the volume of the grating groove, and *V*_2_ is the deformation volume of PDMS after pressurization. Based on this, the change in material deformation on the grating surface under repeated pressure filling was further simulated to obtain the final particle filling rate.

While silicon and PDMS can be found directly in the COMSOL Multiphysics materials library, the particle model is not available. Therefore, we used a porous model instead and obtained the relevant parameters of the porous model through experimentation.

In order to obtain the parameters of the particle model, we made a groove on the surface of a silicon wafer using SU8 2100 photoresist with an area of 10 mm × 20 mm. The particles were deposited in the groove and a hard material, such as a silicon wafer, was placed directly on the surface of the particles. Different pressures were applied to the material surface with an embossing machine in order to obtain the corresponding height of particle variation. The relationship between the variation of particle stacking density when different pressures were applied, i.e., particle filling rate, could be obtained.

## 3. Results and Discussion

### 3.1. Effect of Pressure

After deposition of the particles, the excess particles were removed from the grating surface and the mass of Gd_2_O_3_ particles filled into the grating grooves were obtained via the weighing method. Dividing the mass of the particles by the total mass of the fully filled Gd_2_O_3_ material at the current volume, we determined that the initial filling rate of the grating was 25%. When using Gd_2_O_3_ material as an absorber, an absorption grating with a depth of 70 µm needed to be fabricated in order to fully absorb the neutron at a wavelength of 4.1 Å. To facilitate this study, the grating groove depth was generally set to 70 µm for subsequent experiments and simulations.

The deformation of the PDMS in the grating groove, i.e., the height variation of the particles and the pressure on the particle surface, was obtained by inputting the porous model into the simulation and applying a boundary load to the PDMS surface, as shown in Figure 2b. From the simulation results, when applying a pressure of 0.5 × 10^7^ Pa on the PDMS material on a grating structure with a groove width of 70 µm, we were able to obtain a pressure of 0.3 × 10^7^ Pa on the surface of the particles and a deformation variable of 12 µm for the PDMS. The plot in Figure 2c illustrates the variation in particle height resulting from a pressurized filling experiment conducted on a grating with a 70 µm groove width made of SU8 2100 photoresist. The experiment was performed using a PDMS with Young’s modulus of 7.5 × 10^5^ Pa and a pressure of 0.5 × 10^7^ Pa applied to it. The deformation variable obtained from this experiment is in excellent accordance with Figure 2b. It could also further simulate the change in material deformation under the repeated pressurized filling to obtain the final particle filling rate.

Figure 3a shows the relationship between particle filling rate and pressure in experiments and simulations, where the simulation results were obtained by applying pressure onto the PDMS surface above a grating with a groove width of 20 µm, and the experimental results were obtained based on previous experiments involving porous particle models applying pressure onto the particle surface. It can be observed that, as the pressure applied to the surface of the PDMS increased, the filling rate increased correspondingly, which is consistent with the trend observed in the experimental results. However, it should be noted that the filling rate obtained by applying the same force on the surface of the PDMS was smaller than the experimental results. Simulations enabled us to obtain the pressure transmitted to the particle surface; it was smaller than that applied to the PDMS surface due to deformation. Comparing the relationship between the filling rate and the pressure on the particle surface, both the experimental and simulation results were consistent, indicating that the filling rate was determined by the pressure applied to the particle surface. As the pressure on the particle surface increased, the filling rate gradually increased. When the pressure on the surface of the particles reached 4 × 10^7^ Pa, the filling rate increased significantly from 25% to 55%. Based on the pressurized filling method, it was possible to significantly reduce the aspect ratio of the grating, thus reducing the difficulty of producing the grating.

Figure 3b shows the actual experimental and simulated filling rate results for a silicon grating with a period of 40 µm, duty cycle of 0.5, and area of 60 mm × 60 mm, where the filling rate increased from 25% to 32%. Limited by the equipment and the area of grating, a pressure of 0.7 × 10^7^ Pa was chosen for the pressurization experiments. From Figure 3b, it can be observed that the experimental results of repetitive filling were matched with the simulated results. It can also be seen that the particle filling rate increased fastest at the first pressurized filling and converged quickly, while the filling rate increased slowly with successive pressurized filling. The pressure on the particle surface after each pressurization were obtained from the simulation, as shown by the red dotted line in Figure 3b. Combining the results of points A and B in Figure 3a, it can be seen that when the pressure on the surface of PDMS material was 0.7 × 10^7^ Pa, the particle surface pressure in the simulation was 0.35 × 10^7^ Pa and the filling rate was 32%, which is consistent with the result in Figure 3b when the particle surface pressure was 0.35 × 10^7^ Pa and the filling rate was about 32%. It can be observed that, by applying a fixed pressure to the surface of the grating material, the pressure on the particle surface gradually increased with repeated pressure filling, and the filling rate also increased accordingly. This result is consistent with the relationship between particle surface pressure and filling rate shown in Figure 3a, further indicating that the filling rate is determined by the pressure on the particle surface.

Uniformity is a critical issue to consider when fabricating gratings using the particle filling method, particularly for large-sized gratings where non-uniform filling is more prominent. The non-uniform filling rate during the pressurized filling process was analyzed via a simulation, as shown in Figure 3c, where the initial filling rates in the grating grooves were 20%, 25% and 29%, respectively. It can be observed that, when the initial particle filling rate in the grating groove was different, applying the same pressure produced an adaptive deformation, where the smaller the filling rate, the larger the deformation of PDMS. It can be seen from the simulations that each grating groove can be considered as individually filled and not affecting each other. Meanwhile, the filling rate in the grating grooves increased rapidly after a single pressurized filling, and then quickly approached the same filling rate. Therefore, the pressurized filling method was adapted to the fabrication of large-size absorption gratings to achieve uniform filling.

### 3.2. Effect of Structure and Material

The filling rate is related to the arrangement of the particles. From Figure 3a, it can be observed that the particle filling rate was determined by the pressure on the particle surface. The pressure applied on the surface of PDMS transferred to the particle surface was influenced by the grating groove width and the material of the grating surface. Therefore, the filling rate may also be related to the grating groove width and the surface material.

Figure 4 simulates the variation of the filling rate with the grating groove width obtained by means of repeated filling under a different Young’s modulus of PDMS material. In order to study the effect of different materials on the filling rate and to simplify the simulation, the PDMS’s Young’s modulus was directly varied to perform the study, since the PDMS would have a different Young’s modulus for the different rates of curing agents. The specific parameters were as follows: the pressure applied to the soft material PDMS on the grating surface was 4 × 10^7^ Pa, the grating groove width increased from 3 μm to 20 μm, and the Young’s modulus of PDMS were 5 × 10^5^ Pa, 7.5 × 10^5^ Pa, and 1.5 × 10^6^ Pa. From Figure 4, it can be seen that as the grating groove width increased and the Young’s modulus of PDMS decreased, the filling rate achieved by repeated filling gradually increased, and the effect of the Young’s modulus on the filling was not significant. Moreover, it can be observed that the increase in filling rate using the pressurized filling method was not significant when the grating width was relatively small. Therefore, we have determined that, in order to improve the filling rate of the small-period grating, the surface pressure applied can be increased. However, obtaining high pressure for large-sized gratings is challenging. Thus, splicing and pressurizing small areas can be utilized to generate high pressure, enabling the fabrication of large-sized absorption gratings. With an overlap in the splicing and pressurization process, the pressurized filling method ensures a consistent filling rate without introducing uniformity problems because it is an adaptive filling process, thereby allowing the fabrication of large-size and uniform absorption gratings.

### 3.3. Process Optimization

As can be seen from Figure 3, repeated pressurized fillings were often required many times, resulting in a long production time and low efficiency. Therefore, as a solution, we propose depositing excessive particles on the grating surface, and then applying pressure on the deformed material surface. In order to facilitate the simulation, the model is simplified, as shown in Figure 5. When applying downward pressure to the material, the particles on the surface of the grating lines are compacted, which can be simplified to the grating lines, resulting in an increase in the depth of the grating grooves. Subsequently, the grating surface deformation material deforms between the grating grooves and presses the particles in the grooves. In order to obtain an absorption grating, it is finally necessary to remove the excess particles from the grating surface. The particle layer on the grating surface is immersed with ethanol, followed by the careful removal of particles using a scraper to ensure no residual particles remain on the grating surface.

The effectiveness of the proposed optimized process was investigated through experiments and simulations by depositing excess particles on the grating surface and applying pressure onto the PDMS material surface. The grating structure with a 40 μm period and 20 μm groove width was simulated. The results of a single optimized filling are shown in Figure 6, demonstrating a significantly higher filling rate than that achieved by a single pressurized filling.

After depositing particles on the surface of the grating structure, it was determined via simulation that the particle filling rate could reach about 31% after one filling by applying a pressure of 0.7 × 10^7^ Pa on the material surface. Compared with point C in Figure 3b, this could be reached with the effect of six repeated fillings. The experimental results also demonstrated that a single filling of excess particles deposited on a grating could achieve a filling rate of 30.5%, which is consistent with the simulation results. During particle deposition, non-uniformity remains a concern, as can be seen in Figure 3c, where after a single pressurization, the filling rate was close to its maximum value. To be on the safe side, two optimization pressurizations can also be used to ensure a consistent filling rate.

The optimized process enables the production of a neutron-absorbing grating that can fully absorb neutron beams with a single pressurized filling, thereby greatly improving the filling efficiency. After pressure is applied, the excess layer of particles on the surface will be pressed down as a whole and the filling rate in the groove will increase, meaning that the optimized filling process is even more effective for the fabrication of small period gratings, making the pressurized filling process adaptable to the production of smaller-period gratings and increasing the scope of application.

## 4. Conclusions

In this study, we have introduced a pressurized filling method to enhance the filling rate of absorption gratings fabricated using the particle filling method. The results show that the filling rate is determined by the pressure, and the pressurized filling method can greatly improve the filling rate of the grating. Based on the simulation of the influence of pressure, grating groove width, and surface material on the filling rate, it can be concluded that the filling rate gradually increases with an increase in pressure and grating groove width. Moreover, the pressurized filling process is adaptable and allows for the fabrication of large-size gratings. To address the issue of low efficiency in fabricating the absorption grating using the pressurized filling method, we have proposed a process optimization approach that yields significantly higher efficiency. Therefore, our research provides a promising and practical solution for improving the filling rate and efficiency of the fabrication of absorption gratings.

## Figures and Tables

**Figure 1 micromachines-14-01016-f001:**
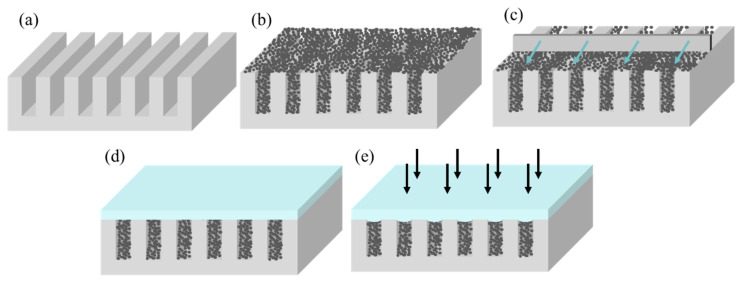
The fabrication process of neutron absorption grating. (**a**) Wet etching to obtain silicon gratings; (**b**) Particle deposition; (**c**) Scraping off excess particles; (**d**) Placement of material on the grating surface; (**e**) Pressure applied to the surface.

**Figure 2 micromachines-14-01016-f002:**
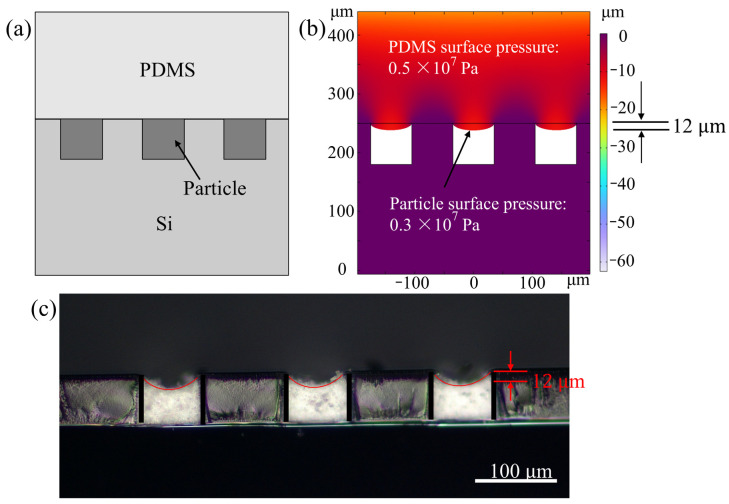
Pressurized filling method. (**a**) Diagram of the simulation model. (**b**) Deformation in simulation. (**c**) Actual experimental particle surface deformation diagram.

**Figure 3 micromachines-14-01016-f003:**
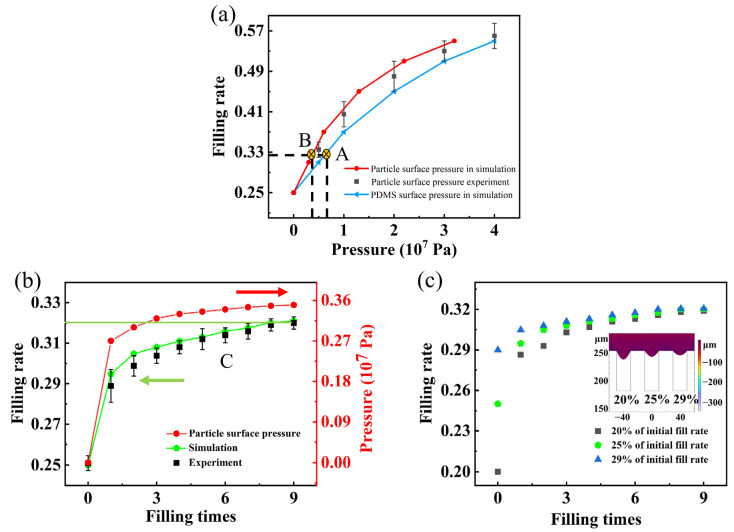
Experiment and simulation. (**a**) Relationship between pressure and particle filling rate, where points A and B indicate that the fill rate reaches 32% when the PDMS surface pressure is 0.7 × 10^7^ Pa and the particle surface pressure is 0.35 × 10^7^ Pa. (**b**) Comparison of experimental and simulated repetitive filling: The green solid line represents the maximum filling rate achieved during repetitive pressurization, while the arrows point to the y-axis corresponding to the different data. Point C denotes the filling rate achievable after six repetitions of filling. (**c**) Repetitive filling of gratings with different initial filling rates.

**Figure 4 micromachines-14-01016-f004:**
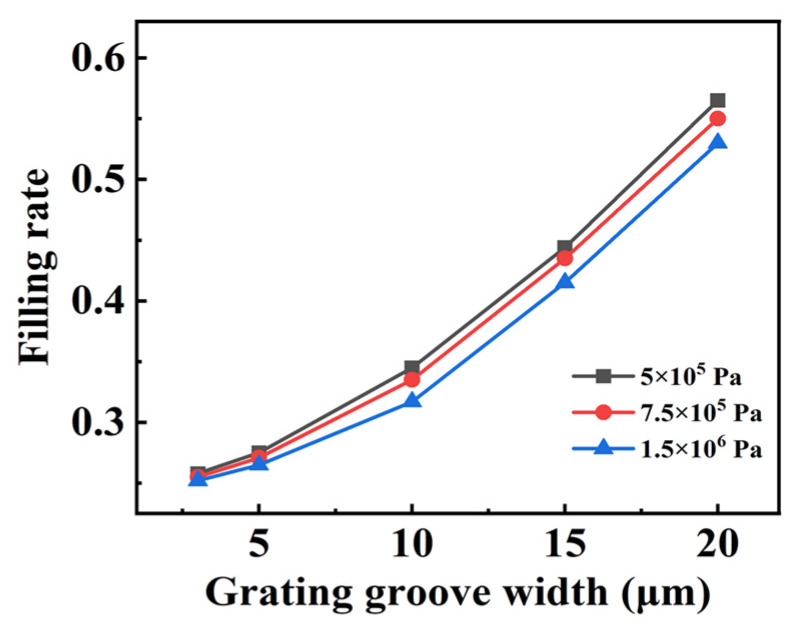
Variation of filling rate for different grating groove widths and Young’s modulus of PDMS.

**Figure 5 micromachines-14-01016-f005:**
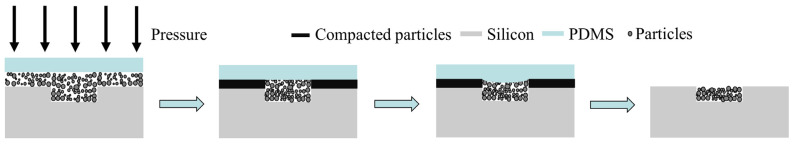
Process optimization diagram.

**Figure 6 micromachines-14-01016-f006:**
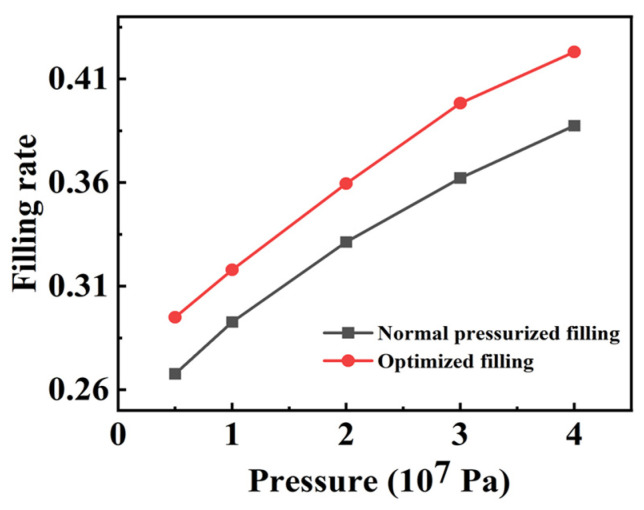
Comparison of filling rate after optimizing filling and normal pressurized filling.

**Table 1 micromachines-14-01016-t001:** Grating structures and experimental parameters.

	Period (μm)	Duty Cycle	Groove Depth (μm)	Grating Area (mm^2^)	Pressure (10^7^ Pa)
Grating (a)	140	0.5	70	60 × 60	0.5
Grating (b)	40	0.5	70	60 × 60	0.7

## Data Availability

The data in the manuscript can be obtained from the corresponding author.

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
