# Peer review of "Simulation and Experimental Validation of a Pressurized Filling Method for Neutron Absorption Grating"

_micromachines, 2023, doi:10.3390/mi14051016_

Round 1

Reviewer 1 Report

First of all, I appreciate chance to review the interesting work, and enjoy reading manuscript as much as that. Authors present simulation/experiment work of fabrication method with particle pressurized filling for neutron absorption grating. This work has an identity as research in terms of compliment for possible disadvantages of previously reported fabrication methods using Gd-based particles. I recommended that the manuscript should be corrected as to concerned comments and questions for clarifying work and helping reader understand. I believe if author can clarify all of points, the manuscript can be published at MDPI.

1)      Title can’t focus on the work. The main is simulation of pressurized filling and comparison with experiment, especially for filling rate. Besides, the manuscript doesn’t contain details of fabrication. I suggest a title in the lines of: “Simulation and experimental validation of fabrication design of pressurized filling method for neutron absorption grating”. Otherwise, author find a new title.

2)      “Consequently, various fabrication processes have been developed…”: Please split the references of 10-18 for each fabrication process authors mention.

3)      Section 2.2 Experiment is too general and ambiguous.

a.       Please insert the table of experiments authors conducted, not as mentioned in text. The parameters, such as grating depth, width, pressure, # of pressure, etc.

b.       “Next, gadolinium oxide particles with a size range of 1-3 μm were dispersed in ethanol solution”: Please mention more detail of preparation of Gd particles.

c.       Please describe how gratings were measured.

d.       Please refer product/manufacturer were used in experiment. Gd particles, ultrasonic machine, PDMS, embossing machine, etc.   

4)      Please add name of materials in Figure 2(a) and increase the letter size in Figure 2(b).

5)      Please mention conditions and parameters of result of pressurized-filled grating in Figure 2(c), which are not known in manuscript. # of pressure, modulus of PDMS, etc. With this lack of explanation, reader may wonder 12 micron is a specific constant for pressurized filling methods regardless of conditions. Also, mention the description the photo captured.

6)      Please mention the reason for Gd2O3 not Gd(OH)3, and reason for choosing 4.1 Angstrom.

7)      In Figure 3, please display marks for measurements and lines for simulations. Figure 3(b) can have different colored-axis for clarifying data in single plot.

8)      Please give physical detail of “filling rate” in experiment and simulation. It should be explained because it is a quantitative parameter goes through the manuscript. Equation can give intuitive understanding.

9)      “Combining the results of points A and B in Figure 3(a), it can be seen that when the pressure on the surface of PDMS material is 0.7×107 N/m2, the particle surface pressure in the simulation is 0.35×107 N/m2 and the filling rate is 32%, which is consistent with the results in Figure 3(b).”: Please correct this sentence to clearly explain  what the consistency of results btw (a) and (b) means.

10)   “Figure 4 simulates the variation of the filling rate with the grating groove width obtained by repeated filling under different modulus of PDMS material.”: Young’s modulus or elastic modulus are recommended.

11)   In 3.3 Process optimization, please discuss neutron absorption in compacted particle-based layer. The layer add extra height for particles, but this layer finally give reduced neutron beam profile. Plus, here is my curiosity of difference in density btw compacted particle layer and particle filled structure.

12)   Change the first letter of legend to have a capital.

Reviewer 2 Report

The authors report their work on the fabrication of a neutron absorption grating using a particle-pressurized filling method. The authors presented both the simulation and experimental results of this method, and data acquired from the experiment were analyzed and compared with the simulation. The design of the experiment seems to be complete but lacks rigorousness. Therefore, the referee suggests the manuscript be reconsidered after the following concerns have been addressed:

1. In the introduction part, the authors mentioned that it is required for the absorption gratings to have a thickness higher than 11 μm to provide a high enough attenuation coefficient for the neutron beam that has a wavelength of 4.1 Å, which is not necessarily the truth. It is clear that in a neutron imaging system, the higher attenuation, the better performance. According to the literature, the attenuation length (1/e transmission) for the said neutron beam is about 3 μm, and 11 μm is just the maximum number that ref. 9 can achieve. Make sure these two numbers are not mixed.

2. The authors mentioned that ‘Accordingly, an increase in the aspect ratio of the grating will greatly increase the difficulty of fabrication.’. When the aspect ratios of the silicon templates in this work are well below 5:1, and silicon templates that have much higher aspect ratios (>50:1) can be fabricated with a plasma etching method (doi: 10.3390/mi11090864), then why does this remain an issue? Please explain in the introduction part.

3. In subsection 2.2, it is mentioned that the particle/ethanol ratio is 100:1, which I believe is the ratio of their weight. Please confirm this. if yes, please clarify in the manuscript.

4. In subsection 2.2, the authors claimed that they ‘Apply pressure uniformly’. How was this guaranteed? Which method/tool did they use to apply the pressure? What is the size of the pressure head?

5. In subsection 2.3, a porous particle model was adopted to simplify the simulation. While the porosity and pore size are not mentioned, a ‘particle packing density’ was experimentally measured. Does it mean that the porosity and pore size are irrelevant to the simulation process, or can be waived by introducing the ‘particle packing density’?

6. At the beginning of subsection 3.1, the authors discussed how the filling rate is calculated in this manuscript, which is ‘by weighing the particles before and after deposition by 25%’. This sentence is a little bit confusing, please rephrase. Where does this 25% come from? And also, I noticed that the excess particles were removed in step 3 (Fig.1c), so the weighing happened before or after this step? Could this step affect the filling rate?

7. Page 4/9 Line 153, the authors describe the duty cycle with ‘1:1’, while the community normally uses ‘0.5’ to describe a grating structure that has a trench width that equals half of the grating period. Please revise.

8. The authors use the times that the pressure was applied to the particles to quantify the pressurization process, which seems to be not very rigorous. How long was each pressing? For example, does a 5 s pressing make any difference if compared with 5 times of 1 s pressing?

9. There is a solid green line in Fig.3b, what does it represent? Please explain in its caption.

10. Different pressure units were used throughout the manuscript (e.g. kPa, N/m2). It is strongly recommended to be consistent or include a conversion.

11. The filling rate is calculated based on a strong presumption that the filling is uniform over the entire grating area. But it will be a more sound statement by evaluating the uniformity of their grating more professionally (doi: 10.1063/1.5047055). 

English should be checked, ideally by a native speaker.

Round 2

Reviewer 1 Report

I appreciate author worked for revision with accepting all comments and advice. The manuscript is ready for publish with minor correction. The last comment corrects the sentence (lines 271-272) below by adding how or what to do.

"In order to obtain an absorption grating, it is finally necessary to remove the excess particles from the grating surface to ensure that no particles remain on the grating surface.”

Reviewer 2 Report

In point 1, the authors replied that 1/e attenuation is the 'optimal' condition, which is not correct, 1/e attenuation is just the 'minimal' requirement. Other than this, all the concerns have been addressed and can be considered for publication.

English has been improved.
